# Metallization and molecular dissociation of dense fluid nitrogen

Shuqing Jiang[1,2], Nicholas Holtgrewe[2,3,6], Sergey S. Lobanov [2,4,7], Fuhai Su[1,2], Mohammad F. Mahmood[3], R. Stewart McWilliams[2,5] & Alexander F. Goncharov [1,2]

Diatomic nitrogen is an archetypal molecular system known for its exceptional stability and complex behavior at high pressures and temperatures, including rich solid polymorphism, formation of energetic states, and an insulator-to-metal transformation coupled to a change in chemical bonding. However, the thermobaric conditions of the fluid molecular–polymer phase boundary and associated metallization have not been experimentally established. Here, by applying dynamic laser heating of compressed nitrogen and using fast optical spectroscopy to study electronic properties, we observe a transformation from insulating (molecular) to conducting dense fluid nitrogen at temperatures that decrease with pressure and establish that metallization, and presumably fluid polymerization, occurs above 125 GPa at 2500 K. Our observations create a better understanding of the interplay between molecular dissociation, melting, and metallization revealing features that are common in simple molecular systems.

[1] Key Laboratory of Materials Physics, Institute of Solid State Physics, Chinese Academy of Sciences, 230031 Hefei, Anhui, China. [2] Geophysical Laboratory, Carnegie Institution of Washington, Washington, DC 20015, USA. [3] Department of Mathematics, Howard University, 2400 Sixth Street NW, Washington, DC 20059, USA. [4] Department of Geosciences, Stony Brook University, Stony Brook, NY 11790, USA. [5] School of Physics and Astronomy and Centre for Science at Extreme Conditions, University of Edinburgh, Peter Guthrie Tait Road, Edinburgh EH9 3FD, UK. [6] Present address: Center for Advanced Radiation Sources, University of Chicago, Chicago, IL 60637, USA. [7] Present address: GFZ German Research Center for Geosciences, Section 4.3, Telegrafenberg, 14473 Potsdam, Germany. Correspondence and requests for materials should be addressed to R.S.M. (email: rs.mcwilliams@ed.ac.uk) or to A.F.G. (email: alex@issp.ac.cn)

The physical and chemical properties of pure nitrogen at high pressures ($P$) and temperatures ($T$) including the stability of the molecular state and related formation of electrically conducting phases offer fundamental constraints on how simple molecular systems related to energetic materials respond to extremes. Moreover, these properties are central for understanding how nitrogen, the primary constituent of Earth's atmosphere, behaves in the deep interiors of planets, where it can appear as a molecule due to ammonia dissociation[1], in subduction zones at oxidizing conditions[2], or as an impurity in iron-rich cores at reduced conditions[3]. Exploring nitrogen under extremes is especially relevant given the challenges of reliably predicting[4] and measuring[5–9] the high $P$–$T$ properties of another important diatomic element, hydrogen. In fact, condensed molecular nitrogen under high $P$ and $T$ behaves similar in many respects to hydrogen: each exhibits melting temperature maxima[10–12] and a fluid insulator-to-metal, molecular-to-nonmolecular transformation that is predicted to be continuous at high temperatures and first order at low temperatures[7, 13, 14] and progressive molecular breakdown in the solid state[10, 15–17]. While such thermodynamic and electronic characteristics are similar in the limit of molecular breakdown into a metallic liquid[13, 18], nitrogen differs from hydrogen in its detailed chemical behavior forming a variety of metastable polynitrogen molecules with a reduced bond order[19] due to its ability to adopt differing bonding types (single to triple).

Solid nitrogen at high pressures shows a rich variety of stable and metastable diatomic (triple-bonded) molecular phases[20–24]. Application of pressure favors the stability of single and double bonds thus promoting molecular dissociation and the formation of energetic polyatomic and polymeric structures[25], including amorphous solid (η)[10, 16, 21, 26, 27], cubic-gauche (cg)[17, 28], and layered (LP)[29, 30] structures, with the onset of dissociation shifting to lower $T$ at higher $P$[27, 31]. Probing fluid nitrogen in single-shock experiments revealed anomalous decreases in volume, temperature, and electrical resistivity at 30–60 GPa and 7000–12,000 K[32], interpreted as signatures of molecular dissociation. In contrast, dynamic multiple compression experiments at ~7000 K reported similar cooling effects at ~90 GPa and a transition to a metallic state above 120 GPa[32, 33], leaving, however, intermediate temperatures of 2000–6000 K unexplored above ~50 GPa. The onset of dissociation with increasing $P$ and $T$ has also been linked to an increase of optical absorption[12, 16, 31, 34, 35], manifesting a connection between dissociation and electronic transformation at optical wavelengths. First principles theoretical calculations[12, 13, 36, 37] predict a first-order transformation from molecular insulating fluid to polymeric conducting liquid ending in a critical point near 75 GPa and 4500 K with a continuous dissociation into a semiconducting atomic liquid at higher $T$ and lower $P$.

The previous gap in experimental data probing nitrogen's states and properties at higher pressure owed to restrictions on dynamic compression pathways to above ~6000 K[32, 33] and static compression heating experiments to below ~2500 K[11, 17, 22, 23, 29, 38] (Fig. 1). Here we access this unexplored domain, critical for understanding the dissociation–metallization transformation of nitrogen, by combining static and dynamic techniques in single-pulse laser heating experiments on precompressed $N_2$ samples in a diamond anvil cell, with in situ time-domain measurements of optical emission, transmission, and reflectance in the visible spectral range (480–750 nm) using a streak camera[6, 39]. We present the experimental observation of a transition line that delineates insulating and conducting fluid nitrogen phases (plasma line) and find the conditions of the insulator-to-metal transition that is associated with a long-sought fluid polymeric nitrogen state. These conditions are in a broad agreement with the previous shock wave experiments albeit our

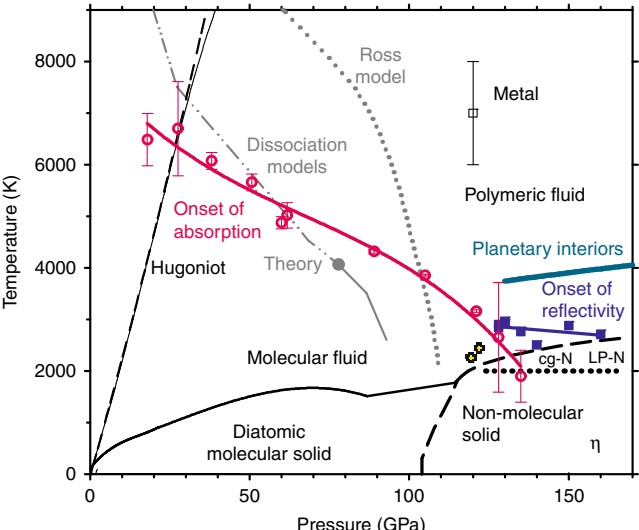

**Fig. 1** Phase diagram of nitrogen at extreme thermobaric conditions. Current measurements of the onset of absorptive states in fluid nitrogen (optical depth ≲10 μm) are presented by magenta circles, while the reflective states are shown by blue squares; solid lines are guides to the eye. Also shown are the single-shock states (Hugoniot) of nitrogen observed experimentally (thin solid black line) and that predicted assuming no chemical dissociation (dashed)[32, 41], metallization in reverberating shock experiments (thin open black square)[33], and fluid–fluid boundaries deduced from double-shock experiments by Ross and Rodgers (gray dotted line)[43] and theoretical calculations (gray solid and dot-dashed lines, indicating regions of first- and second-order transformation, respectively)[36]. The melting line (thick black line) and the domain of nonmolecular solids (thick black dashed line) are from refs. [11, 21], while a dotted black line shows the conditions of formation of crystalline cg-N and LP-N[17, 20, 29]. The results of experiments that recorded crystallization of cg-N on pressure increase[38] are shown by black yellow filled crosses. The turquoise line indicates the conditions in Earth's core[47] and Neptune's deep interior[48]

experiments show that nitrogen reaches the metallic state at much lower temperatures (<2500 vs ~7000 K). The plasma transition is at higher pressures than theoretically predicted posing a challenge for theory.

## Results

**Reaching equilibrium fluid states**. The microseconds (μs) long experiments reported here (Supplementary Fig. 1) are of durations comparable to or greater than shock experiments commonly considered as reaching thermodynamic equilibrium[32] and are sufficient for reaching thermodynamic equilibrium in fluid samples[39]. However, we find this timescale sufficiently short to avoid crystallization of the polymeric solids[17, 29] (e.g., cg-N) in the quenched samples. The $P$–$T$ paths along which we probed the material properties (by varying the heating laser energy at a given nominal pressure) can be considered nearly isobaric, with the maximum added thermal pressure <5 GPa at 3000 K[40] or even smaller given the local nature of the pulsed heating[6].

**Fluid conducting states**. A strong extinction in the transmitted probe light has been detected above a certain threshold laser power; representative data at 121 GPa are shown in Fig. 2. This is clearly seen as a pronounced, transient decrease in intensity of the probe pulses after arrival of the heating pulse. In this regime, we also observe a continuum thermal radiation (Fig. 2a) witnessing the corresponding sample temperature. The transient absorption is the strongest at the times shortly after the highest temperatures

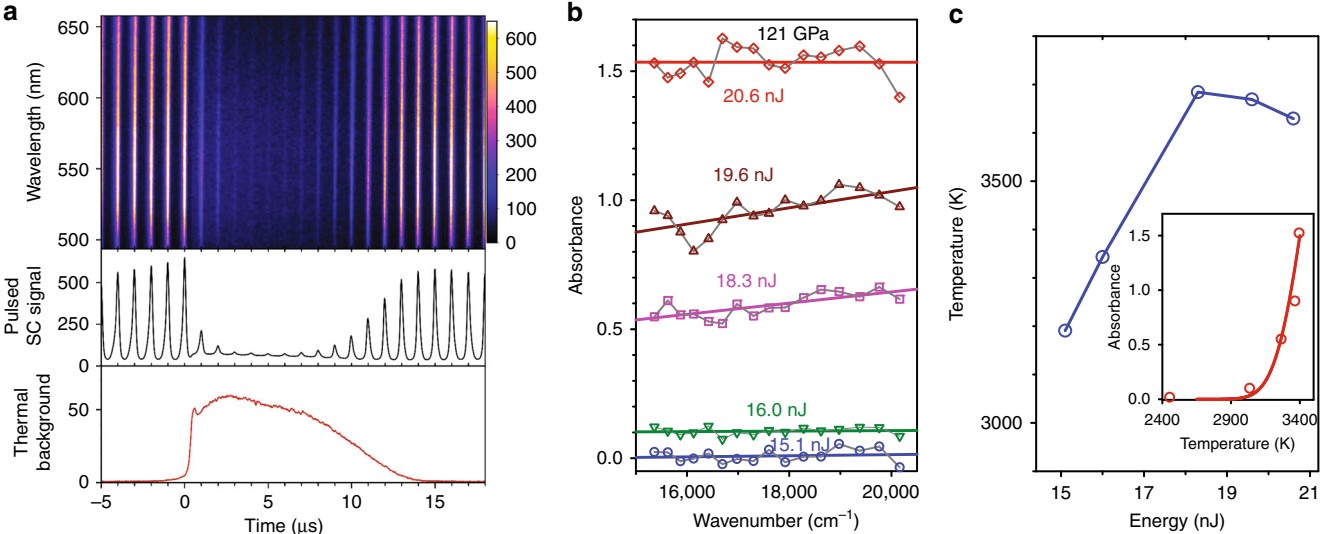

**Fig. 2** Transient absorption measurements in nitrogen at 121 GPa. **a** Upper panel: Spectrogram of transient absorption over 496–651 nm using pulsed supercontinuum broadband probe (SC) during a heat cycle (color scale indicates counts). The vertical lines are the 1 MHz SC probes; the heating laser pulse arrives near time zero, after which nitrogen gradually (within 2–3 μs) becomes hot and the absorptive state is documented through the extinction of the probe; the sample cools back to 300 K after the thirteenth μs, at which time it restores its transparency. Middle panel: the spectrally integrated transmission intensities as a function of time. Bottom panel: spectrally integrated thermal radiation as a function of time. **b** Transient absorption spectra at 121 GPa at various time-averaged laser heating energies; straight lines are guides to the eye. **c** Transient peak radiative temperature as a function of the heating laser energy; the inset shows wavelength averaged absorbance vs peak temperature, with a fit to an error function[39] with a width of 250 K assuming an onset of absorption centered at $T_C = 3400$ K

are recorded at the sample (Supplementary Fig. 2). The spectrograms (Fig. 2a) record the in situ absorption spectra of the sample (Fig. 2b) and corresponding temperature from sample emission (Fig. 2c, Supplementary Figs. 2, 3). The absorption spectra (Fig. 2b) are featureless in the limit of small absorbance but they show an increase with frequency as temperature rises suggesting the presence of the band gap in the near infrared (IR) spectral range as in the case of hydrogen[6]. At these conditions, nitrogen is semiconducting. At even higher temperatures, the spectra again become featureless at the high absorbance limit, suggesting that the bandgap moves further into the IR range or even closes.

As seen in the inset to Fig. 2c, the absorptive state appears over a narrow temperature range of <300 K. Further increase in power does not significantly increase the sample temperature, and the sample remains absorptive. As in the case of hydrogen[6], a plateau in $T$ vs time dependencies occurs once the absorptive state is reached (Supplementary Fig. 2). In our finite element (FE) calculations that model the transient temperature in the sample[39], we used an approximated temperature-dependent absorption coefficient of nitrogen at the laser wavelength, rising near a critical temperature $T_C$ (Fig. 2c). We find that the changes in slope of the dependence of maximum temperature on laser power near $T_C$ (some 150 K above) (Fig. 2c) and in temperature time history can be explained by appearance of the volumetric absorption. Below 121 GPa, the reflectivity spectra did not show any measurable change during heating.

**Liquid metallic states**. Above 121 GPa, we observed a transition to a state with a drastically increased transient reflectivity (Fig. 3). Preceding this at lower laser powers, a regime of elevated absorption is also observed, and the absorption spectra (Fig. 3c) again suggest either the presence of a bandgap in the IR or no bandgap. At the further increase of laser heating powers, sample reflectivity increases gradually (Fig. 3b) and eventually levels off. The reflectance spectra that are observed in these saturated conditions show high values, exceeding several tens of percent,

and a pronounced increase to lower energy (plasma edge), signaling formation of a metallic state. Our reflection spectra, measured from the interface between the metallic and semi-conducting hot nitrogen, can be well fitted using the Drude free electron model (Fig. 3d and Methods section), having two variable parameters: the plasma frequency $\Omega_P$ and the mean free time between the electron collisions $\tau$. The fits yield $\Omega_P = 5.6$ (3) eV and $\tau = 0.9$ (3) fs, which are characteristic of a good metal. The large experimental uncertainty in $\tau$ is due to the narrowness of the spectral range of observations. The electrical direct current (DC) conductivity we obtained in the saturation regime is $\sigma_0 = 5600$ (1700) S/cm in a good agreement with shock wave measurements (≥2000 S/cm)[33] and theoretical predictions (≈3000 S/cm)[35] for the metallic fluid, with the small differences likely due to low-frequency deviation from the Drude model[6, 35, 39]; moreover, the predicted reflectance spectra[35] are only in a slight variance with our measurements. A large experimental error in the value of conductivity includes uncertainty in the correction for absorption of semiconducting nitrogen and in the contribution of bound electrons (Methods section).

## Discussion

All our observations of transient changes in optical properties during heating events are nearly reversible, with small, permanent enhancement of sample absorption and changes to Raman spectra (Supplementary Fig. 4) above 121 GPa; these are related to the formation of an amorphous η state[27, 31]. Transient optical absorption at these extreme $P$–$T$ conditions is not related to formation of cg/LP states as these quench to ambient temperature after heating[17, 31], which was not observed here. The absence of cg or LP nitrogen in our experiments is attributed to short heating times: even ~100 heating cycles (our maximum at each pressure) only yields about 0.5 ms of total heating, much less than that required to crystallize cg- or LP-N in prior studies[17, 20, 29, 31].

Our measurements show that fluid $N_2$ transforms to an absorptive state along a phase boundary with a negative slope

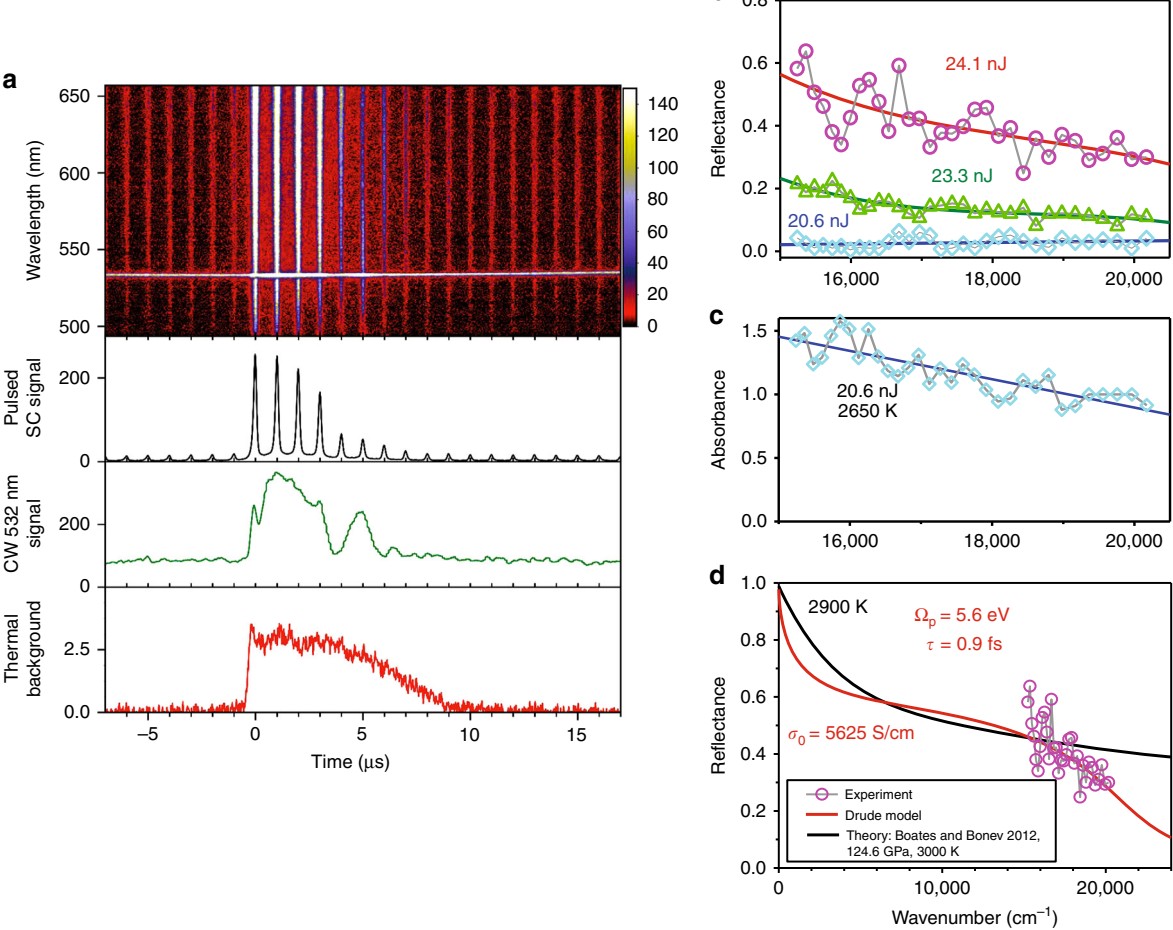

**Fig. 3** Optical transmission and reflectance measurements of nitrogen at 128 GPa through the transformation to a conducting state. **a** Upper panel: Spectrogram of transient reflectivity, which increases drastically in the heated state and returns to its initial state after the ninth μs. Bottom three panels: wavelength-integrated SC pulsed and 532 nm continuous probes and thermal radiation intensities as a function of time, respectively. **b** Transient reflectance spectra as a function of the laser energy picturing the transition into the high reflectance state; the spectra are normalized to the reference spectra of the diamond–N2 interface obtained without heating and determined using the refractive indices of diamond and compressed cold N2 and the transmission of the hot N2 layer (Methods section and Supplementary Figs. 5, 6). **c** Transient absorption spectra corresponding to the lowest energy reflectance spectrum of **b** and used to normalize the reflectance spectra of **b** (Supplementary Fig. 6). **d** An example of the reflectance spectrum along with a Drude fit (Methods section), and theoretical calculations of ref. [35] for polymeric conducting fluid yielding the DC conductivity of $\sigma_0 \approx 3000$ S/cm

(Fig. 1). In the limit of low $P$, this line agrees well with the conditions of anomalous behavior on the shock Hugoniot[32] assigned to molecular dissociation (a departure of the Hugoniot from a normal behavior, Fig. 1)[41]. Meanwhile, Raman measurements of nitrogen shocked to just below the optical transformation boundary, up to 22 GPa and 5000 K[34], find a progressively weakening but still stable diatomic bond, as well as the first signs of weak absorption. Together with our observation of the strong absorption onset at higher temperature and the anomalous cooling effects in dynamic compression[32], we adopt the interpretation that molecular dissociation occurs through a large $P$–$T$ space around the transformation line measured here. The material is semiconducting at these conditions transforming to a conducting atomic plasma only at higher temperatures (e.g., ref. [18]).

Above 121 GPa, nitrogen becomes strongly reflecting in the visible range consistent with the occurrence of the metallic state. The $P$–$T$ line along which this transition occurs has a nearly flat slope in a marked contrast to the steep negative slope of the transition to the absorptive state. The pressure at which nitrogen becomes reflective is very close to the conditions where cg-N starts to form[17], suggesting that the molecular dissociation energy

is close to zero[42]. At these $P$ conditions, the $T$ difference between the appearance of the absorptive and reflective states is very small, suggesting that $T$-induced dissociation is abrupt[18]. In fact, the onset of reflection is close to the conditions where the dense phases of nonmolecular (insulating and semiconducting) solid nitrogen are expected to melt[11, 17, 43], suggesting that the metallization transition may occur upon melting, similar to carbon[44]. We suggest $P_C = 120$ GPa and $T_C = 3000$ K as the critical point delineating continuous and discontinuous molecular dissociation transitions, as predicted theoretically albeit at much lower pressures and higher temperatures[13, 36]. Thus the fluid fully dissociated state of nitrogen occurs above 120 GPa (in agreement with the shock wave data[33] but substantially higher than predicted by theory[12, 35, 37]) and is sufficiently dense to support metallic conductivity[12], which may explain a small negative slope of the transition to a metallic state. Recent X-ray diffraction experiments to just lower temperatures and pressures than the critical point[38] found no evidence for expected liquid–liquid transformations and require that these occur at higher pressure and temperature consistent with our observations (Fig. 1) while generally supporting our assessment that metallization occurs in the atomic state. Note that our metallic fluid N above 125 GPa

was likely produced by melting of metastable bulk η nitrogen, while melting and metallization from cg-N (which was not observed in this work for kinetic reasons) at these pressure conditions remains unexplored.

Our results show interesting parallels to the metallization of hydrogen in a fluid state (e.g., refs. [4, 7]). Even though $N_2$ and $H_2$ molecules are different in chemical bonding, we confirm that major features of their phase diagrams related to molecular dissociation and metallization are very similar. Our observations of two distinct boundaries for the semiconducting and metallic fluid states, and a higher pressure for sudden metallization than theoretically predicted, resonate with recent reports on hydrogen isotopes[6, 7]. As observed here in nitrogen, the pressures at which dissociated fluid metallic states occur in hydrogen are close to the conditions of the formation of solid phases with the features of nonmolecular chemical bonding such as in phases IV and V above 200 GPa[10, 15, 45]. With respect to planetary interiors, although molecular nitrogen in an enriched state remains stable throughout the conditions of Earth's mantle, it dissociates into a fluid metal at the $P$–$T$ conditions of the Earth's core (Fig. 1). This could affect nitrogen's geochemical preference for the Earth's metallic outer core and hence core light element content and Earth's deep nitrogen cycle[2].

## Methods

**Dynamic laser heating and optical probes in the diamond anvil cell**. Our time-resolved single-pulse laser heating diamond anvil cell experiments combine measurements of optical emission, transmission, and reflectance spectroscopy in the visible spectral range (480–750 nm) using a streak camera, as has been described in our previous publications[6, 39] (Supplementary Fig. 1). Nitrogen was loaded in a high-pressure cavity along with a metallic (Ir) suspended foil (coupler), which has one or several cylindrical holes of 5–8 μm in diameter. The sample is conductively heated locally via the energy transport of a near IR (1064 nm) laser focused down to approximately 10 μm (normally from both sides) and absorbed by the rim of the coupler surrounding the hole (or by a flat coupler surface in a few experiments). The laser pulses of 4–10 μs duration are sufficiently long to transfer heat to the nitrogen sample in the hole of the coupler creating a localized heated state of several μm in linear dimensions and a few μs long as determined in our FE calculations (Supplementary Fig. 2). The optical spectroscopic probes, aligned to the heated spot, were used in a confocal geometry suppressing spurious probe signals. Transient transmittance and reflectivity were obtained using pulsed broadband supercontinuum (1 MHz, 150 ps, 480–720 nm) and (occasionally) continuous laser (532 nm) probes having focal spots of approximately 6 μm in diameter (Supplementary Fig. 1) that are spatially filtered with a confocal aperture of some 50% larger in diameter.

Time-resolved (with the resolution down to 0.5 μs) sample temperature was obtained from fitting thermal radiation spectra emitted by the coupler and hot sample to a Planck function (Supplementary Fig. 3). These were normally determined in a separate experiment with identical heating without probing and integrated over a number of laser heating events (5–20) to improve signal-to-noise. The measured temperature should be treated cautiously as the thermal radiation measured represent a sum of contributions from the Ir coupler and the sample. Normally one expects that coupler has higher temperature than the sample and emits more because of difference in emissivity. However, the sample emissivity changes substantially once it becomes absorptive, suggesting that the measured thermal emission in this regime characterizes the sample temperature. Additionally, FE calculations[6, 39, 46] have been used to model the temperature distribution in the high-pressure cavity to estimate the necessary corrections.

At each nominal pressure, temperature was increased stepwise with an increase of the heating laser power. Pressure was measured before and after the heating cycles using Raman spectra of the nitrogen vibrons and stressed diamond edge (Supplementary Fig. 4). Pressure was found to remain essentially constant (within <3 GPa) during heating. No correction for the thermal pressure has been used.

**Transient optical data reduction**. To determine the transient reflectance values of conducting nitrogen at extreme $P$–$T$ conditions, the following procedure has been adopted. The reflectance of the outside diamond–air interface has been a natural reflectivity standard for many experiments of such kind. However, in the majority of experiments reported here this reference reflectivity has not been measured for two reasons. First, it is normally too strong and without the unwanted attenuation can damage the streak camera detector, and second, it requires an additional correction for the absorption of diamonds and change in geometrical factors, which is normally overlooked. However, these measurements have been performed in two experiments and yielded generally consistent results.

In the majority of the experiments, we have used the reflectivity (or transmission in the associated experiments) of the diamond–sample interface at room temperature measured just before the laser heating shot as the reference (Supplementary Fig. 6). The reflectivity of the diamond–nitrogen interface is determined as $R_{nd} = (n_N - n_D)^2/(n_N + n_D)^2$, where $n_N$ and $n_D$ are the refractive indices of nitrogen and diamond, respectively, in the high-pressure chamber. While we assumed that the refractive index of diamond is the same as at ambient pressure, we have performed a separate experiment in molecular nitrogen at 75 GPa to determine its refractive index (Supplementary Fig. 5).

Furthermore, at high temperatures due to large temperature gradients, the conducting state of nitrogen occurs only inside a small hole of the coupler (Supplementary Fig. 6) or immediately next to the coupler flat surface (if the hole is not used). Thus there is an additional interface between the reflective state and hot but untransformed (semiconducting) nitrogen. Because temperature is the continuous function of the coordinate in the experimental cavity, the conducting reflective nitrogen is in contact with the absorptive nitrogen, the transmission of which has been determined in the preceding high temperature shot with a slightly smaller laser energy but almost the same recorded temperature. It has been assumed that the reflectivity of the conducting highly reflective nitrogen was attenuated by a strong absorption of nitrogen that resides at smaller $T$ and has not been transformed.

**Drude model**. Reflectivity spectra are well fitted by a Drude model having conductivity of the form $\sigma = \sigma_0(1-i\omega\tau)^{-1}$, where $\omega$ is the angular frequency. The DC conductivity $\sigma_0 = \Omega_p^2\tau\varepsilon_0$, in which $\Omega_p$ is the plasma frequency, $\tau$ the scattering time, and $\varepsilon_0$ the permittivity of free space. The dielectric constant is $\varepsilon^* = \varepsilon_b + i\sigma$ $(\omega\varepsilon_0)^{-1}$ where $\varepsilon_b$ is the bound electron contribution to the dielectric constant. A range of $\varepsilon_b$ from 1 to $n_N^2$ are examined when assessing uncertainty; we generally found $\varepsilon_b = n_N^2$ provided a better fit to the data. The corresponding index of refraction is $n^* = \sqrt{\varepsilon^*}$. The reflectivity of the interface between the metallic and semiconducting nitrogen is modeled as $R = |(n^* - n_N)/(n^* + n_N)|^2$, where the refractive index of the semiconducting state is assumed to be the same as for molecular nitrogen determined in our separate experiment (Supplementary Fig. 5); use of this expression is strictly valid for a sharp interface between the two states occurring at a particular temperature in the cell cavity (Supplementary Fig. 6), as is expected for metallization based on our observations.

**Data availability**. All relevant numerical data are available from the authors upon request.

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

## Acknowledgements

This work was supported by the NSF (grant numbers DMR-1039807, EAR-1015239, EAR-1520648, EAR/IF-1128867, and EAR-1763287), the Army Research Office (grant 56122-CH-H), the Deep Carbon Observatory, the Carnegie Institution of Washington, the National Natural Science Foundation of China (grant numbers 11504382, 21473211, 11674330, and 51727806), the Chinese Academy of Science (grant number YZ201524), a Science Challenge Project No. TZ201601, an EPSRC First Grant (number EP/P024513/1), and the British Council Researcher Links Programme. A.F.G. was partially supportedby Chinese Academy of Sciences Visiting Professorship for Senior InternationalScientists Grant 2011T2J20 and Recruitment Program of ForeignExperts.

## Author contributions

R.S.M. and A.F.G. designed this study. S.J., R.S.M., N.H., S.L., F.S and A.F.G. conducted experiments. S.J., R.S.M., N.H. and A.F.G. reduced the data. A.F.G. performed finite-element modeling. All authors contributed to the data analysis and wrote the manuscript.

## Additional information

**Competing interests:** The authors declare no competing interests.

