## [Peer Review File · Nature Communications]

Reviewers' comments:

Reviewer #1 (Remarks to the Author):

After the revision, the manuscript is getting better than the previous version. The paper is publishable if the authors can make further efforts to improve the quality of the work. I think the authors should focus on the understanding of phase diagram of pure nitrogen at high temperature regime, neither the implication on hydrogen nor the dynamo origin of giant planet, as I previously suggested. Below are some suggestions.

- 1, Although molecular fluid/solid nitrogen or hydrogen is thought to be possible to dissociate into atomic phase under compression, stable atomic nitrogen phases (e.g. cg-N, LP-N) in solid do not show the metallic nature. Instead, atomic hydrogen structures become metallic and behave alkali metal like. The authors should provide a more direct connection between the metallization process of nitrogen or hydrogen, or they have to remove this comparison.
- 2, If the authors still hope to emphasize that their finding on metallization of nitrogen is helpful to toward understanding of the dynamo origin in giant planets, e.g. Uranus and Neptune. They should put the P-T lines of NH₃ dissociation above 120 GPa for Uranus and Neptune in Fig. 1 like it does in Ref. 9.
- 3, At page 4, the authors mentioned their measured electrical DC conductivity is in good agreement with previous measurements and theoretical predictions. They might give the number that is measured or predicted in the previous references. Otherwise the readers are hard to compare them with the present measurement.
- 4, The new PRL paper of PRL 119, 235701 (2017) by Weck et al. on the similar topic of liquid nitrogen under high pressure should be discussed and compared. Apparently, they did not observe the metallization though Weck et al did not explore a higher temperature as it does in this work. However, they did observe the solid cg-phase when heating while it is missed by this work. The authors might want to find the reason for the discrepancy.

Reviewer #2 (Remarks to the Author):

This paper describes a new technological development useful for high-pressure research, which can lend to the insights into extreme materials behavior. It also presents new data on liquid nitrogen and its metallization, which could be of interest to high pressure and condensed matter physics communities. The question is if the paper is proper for publication in Nat. Comm. In this regard, considering strong impacts of the nature articles, I would recommend to remove some of highly speculative claims on the planetary implications.

The implication of metallic N₂ at 125 and 2500 K to Neptune's shallow interior and Earth's out core, for example, is really a hypothetical statement. The final sentence of the abstract in particular, which relates this work on nitrogen to hydrogen and then suggests nitrogen as a possible role of metallic nitrogen in the giant planets and the Earth's core, is a "big" stretch of the hypothetical conclusion. Such an extraordinary suggestion requires extraordinary evidence, but the fact remains that it is even unclear if nitrogen is present in the deep planetary interiors and the

Earth' core in any significance. The evidences (both theory or experiments) indicate exactly the opposite, for example, the presence of NH₃ rather than its decomposition products at the suggested PT regime.

Reviewer #3 (Remarks to the Author):

In the revised manuscript and reply on referee's comments authors addressed all questions raised by the referees.

As it is already mentioned in my previous review (and supported by the opinion of other Referees), the manuscript presents interesting results for understanding the physical behavior of molecular systems at extreme conditions. The referee's comments from previous round of review indicate that the manuscript can sparked off a lively dispute and stimulate further experimental and theoretical studies of the nitrogen under extreme conditions. This makes the manuscript even more interesting.

Although I am still cautious concerning the results of optical measurements to prove the metallization, I am agree with authors that presented results contain enough indication of conducting state of condensed hot nitrogen. This result is of definitive interest for a broad audience and the results are good and clear presented in the manuscript. In my opinion, the manuscript is worthy of publication in Nature Communications.

Detailed response to the reviewers' criticism

Reviewers' comments:

Reviewer #1 (Remarks to the Author):

After the revision, the manuscript is getting better than the previous version. The paper is publishable if the authors can make further efforts to improve the quality of the work. I think the authors should focus on the understanding of phase diagram of pure nitrogen at high temperature regime, neither the implication on hydrogen nor the dynamo origin of giant planet, as I previously suggested. Below are some suggestions.

We thank the referee for appreciating our work. Following the reviewer suggestion, in the revised document we focus on the high P-T phase diagram of nitrogen deemphasizing other implications

1, Although molecular fluid/solid nitrogen or hydrogen is thought to be possible to dissociate into atomic phase under compression, stable atomic nitrogen phases (e.g. cg-N, LP-N) in solid do not show the metallic nature. Instead, atomic hydrogen structures become metallic and behave alkali metal like. The authors should provide a more direct connection between the metallization process of nitrogen or hydrogen, or they have to remove this comparison.

We thank the referee for pointing this out. The connections between metallization of liquid nitrogen and hydrogen at high pressures is in that it occurs at pressures where solid nonmolecular phases occur at low T, thus at the conditions where the molecular dissociation energy is getting close to zero. It is important to note that these solid non-molecular states are reproducibly observed in experiments to be insulating in both hydrogen and nitrogen. We feel that this conclusion based on the result of our work is quite prominent and well established. We slightly focused the relevant discussion in the revised text. Please also note, that the conditions and character of metallization of liquid and solid hydrogen remain a matter of intense debates, thus making reliable relevant observation in related systems very valuable. The alkali-metal-like atomic hydrogen solid referred to by the reviewer has been theoretically proposed but not definitively observed, and may not even exist (e.g. other theory finds the ultimate atomic ground state to be a fluid).

2, If the authors still hope to emphasize that their finding on metallization of nitrogen is helpful to toward understanding of the dynamo origin in giant planets, e.g. Uranus and Neptune. They should put the P-T lines of NH₃ dissociation above 120 GPa for Uranus and Neptune in Fig. 1 like it does in Ref. 9.

We thank the referee for this comment, and have removed this discussion as several reviewers felt it was too speculative.

3, At page 4, the authors mentioned their measured electrical DC conductivity is in good agreement with previous measurements and theoretical predictions. They might give the number that is measured or predicted in the previous references. Otherwise the readers are hard to compare them with the present measurement.

We thank the referee for this comment. In the revised manuscript we provide the values for electrical conductivity as determined in shock wave experiments and theory, and also quantitatively discuss the agreement, enabling us to add further physical insight.

4, The new PRL paper of PRL 119, 235701 (2017) by Weck et al. on the similar topic of liquid nitrogen under high pressure should be discussed and compared. Apparently, they did not observe the metallization though Weck et al did not explore a higher temperature as it does in this work. However, they did observe the solid cg-phase when heating while it is missed by this work. The authors might want to find the reason for the discrepancy.

We thank the referee for this comment and now discuss this paper (which we become aware of after the initial submission) carefully in our manuscript. Concerning the lack of observation of cg phase in our experiments, we note that we provided our explanation in the original manuscript, but now clarify this issue throughout our revised manuscript and in relation to the Weck article. This is a kinetic effect related to the necessity for cg phase to crystallize through spontaneous nucleation, as established in prior studies. The time scale of our experiments (microseconds) was insufficient for cg nucleation, while it could be possible in the “minutes”-long experiments of Weck.

Reviewer #2 (Remarks to the Author):

This paper describes a new technological development useful for high-pressure research, which can lend to the insights into extreme materials behavior. It also presents new data on liquid nitrogen and its metallization, which could be of interest to high pressure and condensed matter physics communities. The question is if the paper is proper for publication in Nat. Comm. In this regard, considering strong impacts of the nature articles, I would recommend to remove some of highly speculative claims on the planetary implications.

We thank the referee for appreciating our work. Following the reviewer suggestion, in the revised document we focus on the high P-T phase diagram of nitrogen removing or de-emphasizing other implications.

The implication of metallic N₂ at 125 and 2500 K to Neptune’s shallow interior and Earth’s out core, for example, is really a hypothetical statement. The final sentence of the abstract in particular, which

relates this work on nitrogen to hydrogen and then suggests nitrogen as a possible role of metallic nitrogen in the giant planets and the Earth's core, is a "big" stretch of the hypothetical conclusion. Such an extraordinary suggestion requires extraordinary evidence, but the fact remains that it is even unclear if nitrogen is present in the deep planetary interiors and the Earth' core in any significance. The evidences (both theory or experiments) indicate exactly the opposite, for example, the presence of NH₃ rather than its decomposition products at the suggested PT regime.

We agree with the reviewer that the behavior of nitrogen in icy giant planet interiors is controversial, and removed most discussion of this issue, including more speculative claims about the role of nitrogen in conductivity of giant planets interiors.

However, we disagree that nitrogen is not present in Earth's interior in any significance: for example, it is a major impurity of diamond and appears in volcanic gases, and there has been significant study of the role of nitrogen in the deep Earth, both in mantle and core (as our citations clarify). We thus include a brief but limited discussion about the properties of pure nitrogen at conditions of Earth's interior, which we probed for the first time, and discuss the implications with all possible caution in the revised manuscript to stimulate further investigation.

Reviewer #3 (Remarks to the Author):

In the revised manuscript and reply on referee's comments authors addressed all questions raised by the referees.

As it is already mentioned in my previous review (and supported by the opinion of other Referees), the manuscript presents interesting results for understanding the physical behavior of molecular systems at extreme conditions. The referee's comments from previous round of review indicate that the manuscript can sparked off a lively dispute and stimulate further experimental and theoretical studies of the nitrogen under extreme conditions. This makes the manuscript even more interesting.

Although I am still cautious concerning the results of optical measurements to prove the metallization, I am agree with authors that presented results contain enough indication of conducting state of condensed hot nitrogen. This result is of definitive interest for a broad audience and the results are good and clear presented in the manuscript. In my opinion, the manuscript is worthy of publication in Nature Communications.

We thank the referee for appreciating our work.

REVIEWERS' COMMENTS:

Reviewer #1 (Remarks to the Author):

The revision is satisfactory to me. I suggest the publication of the work.

Reviewer #2 (Remarks to the Author):

This revision has improved the quality, especially in focusing on the metallization of nitrogen and removing the speculative discussions on the planetary implications. Therefore, I recommend the paper for publication, providing the error bars for the estimated temperatures presented in Fig. 2c.

Reviewer #3 (Remarks to the Author):

As I have mentioned in the previous round of review, the manuscript of Goncharov et al. is devoted to the exploration of the pressure-temperature phase diagram of an archetypical molecular system, which is a matter of fundamental research interest for a decades. Although the possible impact on planetary interior might be still under debate, the observation of the liquid conductive state at high pressures – high temperatures is of definitive fundamental interest. In the revised manuscript and reply on referee's comments authors addressed all questions raised by the referees. The experimental results are well presented and clearly discussed in the manuscript. In my opinion, the manuscript is worthy of publication in Nature Communications.